# Mitochondrial Surveillance by Cdc48/p97: MAD vs. Membrane Fusion

**DOI:** 10.3390/ijms21186841

**Published:** 2020-09-18

**Authors:** Mafalda Escobar-Henriques, Vincent Anton

**Affiliations:** Institute for Genetics, Cologne Excellence Cluster on Cellular Stress Responses in Aging-Associated Diseases (CECAD), Center for Molecular Medicine Cologne (CMMC), University of Cologne, Joseph-Stelzmann-Straße 26, 50931 Cologne, Germany; vanton1@uni-koeln.de

**Keywords:** Cdc48, p97, VCP, ubiquitin, mitochondria, MAD, fusion, mitofusin, Fzo1, Mfn1/2

## Abstract

Cdc48/p97 is a ring-shaped, ATP-driven hexameric motor, essential for cellular viability. It specifically unfolds and extracts ubiquitylated proteins from membranes or protein complexes, mostly targeting them for proteolytic degradation by the proteasome. Cdc48/p97 is involved in a multitude of cellular processes, reaching from cell cycle regulation to signal transduction, also participating in growth or death decisions. The role of Cdc48/p97 in endoplasmic reticulum-associated degradation (ERAD), where it extracts proteins targeted for degradation from the ER membrane, has been extensively described. Here, we present the roles of Cdc48/p97 in mitochondrial regulation. We discuss mitochondrial quality control surveillance by Cdc48/p97 in mitochondrial-associated degradation (MAD), highlighting the potential pathologic significance thereof. Furthermore, we present the current knowledge of how Cdc48/p97 regulates mitofusin activity in outer membrane fusion and how this may impact on neurodegeneration.

## 1. Introduction

Mitochondria are key cellular players, as biosynthetic hubs, in redox signalling and stress responses, beyond their well-known role in bioenergetics [1]. Indeed, cellular bioenergetics heavily depend on mitochondria, which accommodate the electron transport chain (ETC), where oxidative phosphorylation (OXPHOS) leads to the generation of energy. Mitochondrial organelles possess a double membrane, called outer and inner mitochondrial membrane (OMM and IMM, respectively), which compartmentalize mitochondria into two soluble domains, the matrix and the intermembrane space (IMS). The ETC, located in the IMM, is also the main source of reactive oxygen species (ROS), thus regulating cellular signaling and homeostasis [2]. Mitochondria harbor the citric acid cycle, whose intermediates not only serve as amino acid precursors but also play important roles in cellular pathways, such as thermogenesis, tumorigenesis, stem cell function, and immune responses [3]. Further, by enabling the synthesis of nucleotides, fatty acids, glucose, and heme, mitochondria provide an essential part of the cellular metabolites [1]. Mitochondrial compartimentalization also allows the generation of iron-sulfur clusters, essential inorganic cofactors involved in an immense range of cellular pathways [4,5]. In addition, mitochondria allow the synthesis of the phospholipids cardiolipin, exclusive to mitochondria, and phosphatidylethanolamine [6], which must be transferred to the endoplasmic reticulum (ER). Finally, mitochondria play an essential role in buffering Ca^2+^ overload from the ER and cytoplasm [7]. Misregulation of calcium homeostasis is commonly associated with many diseases including metabolic dysfunctions such as type 2 diabetes and neurodegeneration [8,9].

By serving as a cell death gatekeeper, mitochondria play a central role in cellular quality control and survival decisions. Here, formation of the mitochondrial permeability transition pore and release of specific mitochondrial contents to the cytosol can lead to apoptosis or necroptosis [10,11]. Moreover, mitochondrial signalling regulates immunity and inflammatory responses [8,12,13]. The integrity of the mitochondrial pool itself is regulated on an intra-cellular level via mitophagy, whereby compromised organelles are selectively isolated and cleared by lysosomal degradation [14,15]. Failure in mitochondrial integrity and quality control mechanisms is associated to numerous disorders, ranging from neurodegenerative to muscular, gastrointestinal, and metabolic diseases, mostly prevalent during aging [16,17].

Mitochondria are highly dynamic organelles, constantly adapting to environmental conditions, by balancing fusion events, mediated by mitofusins (Mfn1/2, Marf, Fzo1) and OPA1 (Mgm1), with fission events, mediated by Drp1 (Dnm1) [18]. For example, nutrient overload leads to fragmentation of the mitochondrial network, causing higher energy dissipation, while fasting causes interconnection of the organelle, facilitating more efficient ATP synthesis [19]. Defects in mitochondrial plasticity directly cause numerous diseases, such as Charcot-Marie-Tooth (CMT) type 2A, multiple symmetric lipomatosis (MSL), and dominant optic atrophy (DOA) [20]. Moreover, dynamics’ impairment is associated to Parkinson’s, Huntington’s, metabolic, and cardiac diseases [21].

Mitochondrial communication with the cytoplasm, but also inter-organellar signalling, have major implications in cellular homeostasis [22,23,24]. Despite harboring its own DNA and respective transcription and translation machineries [25], most of the mitochondrial proteome is encoded in the nucleus [26]. Consequently, mitochondrial integrity relies on protein import. In addition, mitochondrial homeostasis is maintained by a set of quality control machineries, namely internal proteases, chaperones, and the ubiquitin proteasome system (UPS) [27,28,29,30]. The UPS is mostly prevalent for proteins resident in the OMM, but also impacts others, e.g., during their import into mitochondria.

Ubiquitin is a major cellular and disease gatekeeper [31,32]. It is covalently attached to lysine residues of target proteins, via an enzymatic cascade comprising E1, E2, and E3 enzymes [33,34]. Ubiquitin chains can be cleaved and regulated by deubiquitylases (DUBs), greatly expanding the possibilities for regulation [35]. Ubiquitin-related events are regulated by a highly conserved ATP-dependent, mechano-chemical motor named Cdc48 in yeast and p97 or VCP in mammals [36,37]. Cdc48/p97 recognizes ubiquitylated client proteins and extracts them from membranes, chromatin, or protein complexes, upon ATP hydrolysis. However, a ubiquitin-independent role of Cdc48/p97 was recently identified, in regulatory complex disassembly and protein phosphatase-1 (PP1) holoenzyme activation [38]. Cdc48/p97 was first discovered in *Saccharomyces. cerevisiae*, as an essential cell cycle component for G2-M transitions (Cell Division Cycle) [39]. Cdc48/p97 belongs to the family of AAA (ATPases associated with diverse cellular activities)-ATPases, which exist in all species from bacteria to humans and often function as essential chaperones, promoting protein folding or unfolding [40]. Mutations in p97 are causative of a multitude of muscular and neural degenerative diseases [41], including amyotrophic lateral sclerosis (ALS), Charcot-Marie-Tooth type 2Y, inclusion body myopathy associated with Paget’s disease of the bone and front-temporal dementia (IBMPFD), and multisystem proteinopathy [42,43]. Further, Cdc48/p97 has been established as a target for the development of cancer therapeutics, highlighting its pathophysiological importance [44,45].

Cdc48/p97 is one of the most abundant cellular proteins, with major roles in protein homeostasis [36,37], also recently proposed to directly depend on the cellular ubiquitin pools [46]. It mostly localizes to the cytoplasm, where it surveys multiple cytosolic and organellar processes, but is also present in the nucleus, where it ensures chromatin integrity, facilitates replication, and modulates nuclear proteostasis [47,48,49]. This varying localization is likely mediated by the cofactors of Cdc48/p97, which determine substrate specificity [50]. Importantly, a fraction of Cdc48/p97 associates with membranes of organelles such as ER, Golgi, mitochondria, and endosomes [51,52,53,54,55,56,57,58]. One of the most extensively studied Cdc48/p97 dependent-pathway is the endoplasmatic reticulum-associated degradation (ERAD), where the AAA-ATPase extracts ubiquitylated proteins from the organellar membrane and facilitates their degradation by the proteasome [59,60,61,62,63]. In analogy to its role in ERAD, Cdc48/p97 and the proteasome assist mitochondrial-associated degradation (MAD) [29,64,65,66,67] (Figure 1).

Mitofusins, OMM proteins playing essential roles in cellular homeostasis [68,69], were the first Cdc48/MAD substrates to be identified [58,70,71,72]. Under stress conditions, Cdc48/p97 extracts ubiquitylated mitofusin from OMM and hands it over for proteasomal degradation, which inhibits mitochondrial fusion and also regulates contact sites between mitochondria and the ER [68,73]. In contrast to MAD, Cdc48/p97 also stabilizes the yeast mitofusin Fzo1 and promotes mitochondrial fusion, e.g., by segregating Fzo1 clusters at fusion sites, thereby recycling it for a new fusion event [74,75,76]. In mammalian cells, further examples of p97-dependent MAD have been proposed, highlighting the complexity of the regulation of mitochondrial dynamics by Cdc48/p97. SCL25A46, a distant homologue of the yeast OMM profusion protein Ugo1 (however, fulfilling an antifusion role) is ubiquitylated by the mitochondrial E3 ligases MULAN/Mul1/Mapl and MARCH5/MITOL [77]. This leads to proteasomal- and p97-dependent degradation of SCL25A46. Strikingly, however, MARCH5 itself is negatively regulated by the fission factors Mff and Drp1, also dependent on p97 [78]. To date, as further detailed here, many mitochondrial proteins regulated by either Cdc48/p97 or its cofactors have been identified [79,80,81]. This implicates Cdc48/p97 in a variety of mitochondrial-linked processes, including protein import and translation surveillance, mitophagy, and apoptosis [82,83,84,85,86,87]. Consistently, Cdc48/p97 mutations are causative of a multitude of mitochondrial abnormalities such as mitochondrial aggregation, increased production of ROS, and reduced ATP synthesis [88,89,90,91,92,93,94,95].

In conclusion, maintenance of the mitochondrial proteome is essential for correct mitochondrial function and cellular survival. Here, we reviewed the roles of Cdc48/97 in mitochondria (Figure 1).

## 2. The Molecular Mechanism of Cdc48/p97

AAA-ATPases serve as proteolytic motors, allowing the recruitment and unfolding of substrates carrying specific degradation signals [96,97]. Typically, they form hexameric rings, where the proteolytic subunits assemble into a cylinder-shaped structure, with a pore at the central axis, forming the catalytic chamber. Here, ATP hydrolysis leads to conformational rearrangements, which facilitate the mechanical force needed for substrate processing [98]. Cdc48/p97 assembles into hexameric rings, each monomer consisting of an N-terminal domain, followed by two ATPase domains, called D1 and D2, that are connected by short linkers, termed N-D1 linker and D1-D2 linker, respectively, ending with a short C-terminal tail [96,99] (Figure 2A). The D1 ring was proposed to be responsible for ubiquitin engagement and the initial unfolding [100]. In turn, translocation seems to largely rely on D2, which pulls the substrate through the central pore upon hydrolysis [101,102,103]. Consistently, larger substrate densities were observed in the D2 than in the D1 ring [99,100]. Furthermore, differences in the substrate-binding residues, present in the catalytic rings, support distinct substrate-binding and hydrolysis capacities between the two subunits [99]. Whereas D2 displays aromatic residues, proposed to facilitate unfolding of tightly folded segments, D1 displays smaller and flexible residues, with poor unfolding potential. Nevertheless, D1 and D2 are interdependent, as the ATPase activity of D2 is affected by the ATPase activity of D1 [104,105,106].

Structural analysis of Cdc48/p97 revealed several conformational changes, depending on the nucleotide state of the ATPase, including rotational, back-and-forth movement between the D1 and D2 rings [107,108]. This was suggested to provide the mechanical force that enables segregase activity. Interestingly, D1 was mostly found to be adenosine diphosphate (ADP)-bound, suggesting that D2 activity is the main driving force for these conformational changes [96,101,109]. Furthermore, nucleotide-dependent opening and closing of the central D2 cavity were observed [96,110,111,112]. Lastly, the N-terminal domain of Cdc48 could undergo lateral movements [113]. However, depending on the nucleotide state, not all subunits of Cdc48 engage the same conformation. Therefore, this movement of the N-domain could happen in an asymmetric and unsynchronized manner. Functionally, this likely goes hand in hand with differential binding of cofactors to Cdc48/p97 [50,96,103].

The substrate specificity of Cdc48/p97 depends on its cofactors, whose number is constantly expanding. They bind to the N- or to the C-terminal domain of Cdc48/p97 and recognize either ubiquitin moieties or other cofactors [50] (Figure 2A). However, several interacting motifs have overlapping binding sites. Depending on their binding domains, the cofactors can be subdivided into several classes. The ones binding via ubiquitin regulatory X (UBX), UBX-like (UBXL), VCP-interacting (VIM), VCP-binding motif (VBM), or SHP box domains recognize the N-terminus of Cdc48/p97. Instead, PNGase/UBA or UBX containing proteins (PUB) and PLAP, Ufd3/Doa1, and Lub1 containing proteins (PUL) are targeted to its C-terminus. Additionally, several noncanonical binding partners have been identified, such as neurofibromin 1 (NF1), which interacts with the two ATPase domains via its leucine-rich repeat domain (LRD). Cofactors rarely bind to all six subunits of the hexamer. Moreover, several cofactors form bipartite modules, such as Ufd1-Npl4, that usually bind two subunits of the hexameric complex, via the UBX domain of Npl4 and the SHP box of Ufd1. This high complexity in exclusiveness and interdependency of cofactor binding to Cdc48/p97 confers its tight regulation of activity and localization.

Recent cryo-EM structures of Cdc48-Shp1 (Ubx1) or of Cdc48-Ufd1-Npl4 in complex with a ubiquitylated substrate provided deeper insight into the unfolding mechanism [99,100]. In the case of Cdc48-Ufd1-Npl4, Ufd1 and Npl4 each bound two from the four last moieties of the ubiquitin chain, consistent with previous observations [100,114] (Figure 2B, Ub_2_ to Ub_5_). Strikingly, the subsequent ubiquitin moiety (Ub_1_) was unfolded and bound to Npl4 and the D1 ring, suggesting that binding to Npl4 and the D1 domains facilitates unfolding independently of ATP hydrolysis at D2. By further pulling Ub_1_ into D2, by its N-terminus, Cdc48 then pulls the surrounding ubiquitins, along with the substrate itself, leaving the three distal ubiquitin moieties (Ub_3_-Ub_5_) in their native state. Surprisingly, the observation that unfolding starts at a proximal ubiquitin moiety (Ub_1_), whereas the distal ubiquitin moieties (Ub_3-5_) stay in their native state bound to the Cdc48 cofactors, suggests that at least two ubiquitin polypeptide chains (Ub_1-2_) are located in the central pore simultaneously. This hypothesis is conceivable due to the large empty space observed in the central cavity of the substrate-bound structures. However, how the selective unfolding and preservation of specific ubiquitin moieties is mediated remains unclear. This initial unfolding of ubiquitin explains the ability of the Cdc48-Ufd1-Npl4 complex to unfold compact globular structures lacking unstructured N- or C-terminal segments [100].

Interestingly, nucleotide binding and substrate engagement of Cdc48 exhibit an architecture similar to the 26S proteasome. For the proteasome, which possesses only one ATPase ring, two ATPase subunits are disengaged, while the other four bind the substrate, thus forming a helical staircase [115]. Consistently, for Cdc48, two ATPase subunits of both D1 and D2 are disengaged, while the other four subunits are bound to ATP. Moreover, structures of Cdc48 in complex with Shp1/Ubx1 displayed an asymmetric configuration of both the D1 and the D2 ring upon substrate processing [99]. Here, five subunits are bound to the substrate, four of which are bound to ATP, one to ADP. This configuration largely depends on the interfaces between the subunits. ATP hydrolysis at one of the substrate-bound subunits allows ADP release at the fifth substrate-bound ATPase. This leads to substrate disengagement of this subunit, to shifting of subunit interactions, and to ATP binding, combined with substrate engagement of the previously disengaged subunit. This dynamic rearrangement of the catalytic chamber enables a so-called hand-over-hand mechanism, via which the substrate is transferred between the ATPase subunits and thereby pulled through the catalytic pores of D1 and D2 [99].

## 3. Cdc48/p97 in Mitochondrial-Associated Degradation (MAD) and Respective Quality Control Roles

In mitochondria, the importance of Cdc48/p97 was initially linked to mitochondrial quality control and cellular survival [11]. It originated from the observation that yeast strains carrying a mutation in Cdc48 presented apoptotic signs [88,116], accumulated polyubiquitylated substrates in mitochondrial extracts [117], and also showed major misregulation of numerous mitochondrial proteins, in apoptotic or aged cells [118]. Importantly, evaluations of different rat organs identified p97 to be part of the mitochondrial proteome, in line with a direct role [119]. Nowadays, in addition to apoptosis, mitochondrial quality control roles of Cdc48/p97 span from mitophagy to mitochondrial-associated translation and import of nuclear-encoded mitochondrial proteins [95,120,121]. Further, p97 was shown to clear damaged mitochondrial proteins upon oxidative stress, regulate Ca^2+^ uptake proteins upon calcium overload, and facilitate the degradation S-nitrosylated mitochondrial proteins [122,123,124]. This highlights the importance of Cdc48/p97 for mitochondrial quality control, by assisting the proteasome in the degradation of many different mitochondrial proteins, via MAD. Depending on the metabolic and/or stress condition, Cdc48 is targeted to specific mitochondrial substrates by different cofactors, enabling fine-tuning of the cellular responses.

### 3.1. Ubx2- and Cdc48-Dependent MAD during Mitochondrial Import

The involvement of Cdc48/p97 in ERAD [54,55], which was first identified a decade after the discovery of the first ERAD substrate [125], is well defined [60,61,62]. Cdc48 recruitment to the ER requires Ubx2, an integral ER membrane protein [126,127,128]. Ubx2 also interacts with membrane-bound ER E3 ubiquitin ligases, including Doa10 and Hrd1, which ubiquitylate both lumenal and membrane-anchored ER proteins [129,130,131]. These ERAD substrates are then recognized by Ubx2, by Cdc48, and by its cofactors Ufd1 and Npl4 and are translocated to the cytosol, where they are degraded by the proteasome [59].

Recently, it was shown that Ubx2 also assists the degradation of OMM proteins [81,87,132,133] in addition to its role in ERAD (Figure 3A). Ubx2 was shown to assist Cdc48-Ufd1-Npl4 in recognizing and extracting proteins bound to the translocase of the OMM (TOM) channel, under mild stress. These are subsequently degraded, a mechanism termed mitochondrial protein translocation-associated degradation (mitoTAD) [87]. In this case, a fraction of Ubx2 was proposed to localize to the OMM, where it binds to the TOM complex, consistent with a direct role. In contrast to other MAD pathways described below, the recruitment of Cdc48-Ufd1-Npl4 did not require the Cdc48 cofactors Doa1 or Vms1. However, simultaneous deletion of UBX2 with other MAD components, like Vms1, causes increased ubiquitylation of proteins associated to the TOM channel and mitochondrial defects, indicating redundant roles of these pathways [87]. Moreover, Ubx2 regulates Fzo1 ubiquitylation and turnover [132,133]. Finally, the degradation of unstable, temperature-sensitive mutant variants of OMM substrates, also requires Ubx2, along with the Cdc48-Ufd1-Npl4 complex and with San1 and Ubr1, main quality control E3 ligases [81] (Figure 3A).

### 3.2. Ufd3/Doa1 and Msp1 in Surveillance of Proteins Present at the OMM

Under basal conditions, the Cdc48 cofactor Ufd3/Doa1 is important for a general OMM quality control mechanism by Cdc48-Ufd1-Npl4, involving the quality control E3 ligase Rsp5 [121] (Figure 3B). Doa1 facilitates the recruitment of the Cdc48-Ufd1-Npl4 complex to OMM-anchored proteins, particularly Fzo1 and Tom70, the AAA-ATPase Msp1 and the ER-mitochondria encounter structure (ERMES) component Mdm34, assisting in their proteasomal degradation [79]. However, in absence of stress, Cdc48-dependent degradation of Fzo1 might be due to tagging artefacts [74]. Additionally, a role of Ufd3/Doa1 for Cdc48-dependent MAD under chronic stress conditions was very recently identified, regulating lifespan control [134]. Interestingly, Rps5 also enables ubiquitylation and proteasomal degradation of Fzo1, in response to age-induced vacuole stress [135] and of Mdm34 and Mdm12, another ERMES component, upon induction of mitophagy (Figure 3B) [134].

Msp1/ATAD1 is itself involved in MAD, together with Cdc48, however, by engaging the ERAD pathway (Figure 3C), firstly, by recognizing and extracting ER tail-anchored (TA) proteins mislocalized to the OMM [80,136]. These mistargeted proteins were proposed to then translocate to the ER, where they are handled as classical ERAD targets [80]. Secondly, mislocalized peroxisomal TA proteins are also extracted by Msp1 [137,138]. Finally, Msp1 resolves proteins clogging the protein import machinery [139], emphasizing the importance of Msp1-dependent translocation.

### 3.3. Roles of Vms1 in Oxidative Stress and Ribosomal Quality Control

The Cdc48 cofactor Vms1, whose binding is mutually exclusive with Ufd1, contains a mitochondrial-targeting domain (MTD) [70,140]. Under basal conditions, this MTD is masked, by being bound to a leucine-rich sequence (LRS) present at the N-terminus of Vms1 (Figure 3D). Upon oxidative stress, the oxidized sterol ergosterol peroxide leads to mitochondrial localization of Vms1, by competing with the Vms1-LRS for binding to the MTD [84,140,141]. Consistently, in mammalian cells, p97 was shown to be essential for clearance of oxidatively damaged mitochondrial proteins [122].

Vms1 is also required for the ribosomal quality control pathway (RQC), a proteostasis pathway, mainly understood for cytoplasmic proteins, that is activated by misfolding of nascent polypeptide chains or by ribosome stalling during translation [142,143,144,145,146,147]. Importantly, it was recently discovered that Vms1 exhibits peptidyl-tRNA hydrolase activity [86,148,149]. Vms1 is recruited to stalled ribosomes and helps to dissociate and release the stalled ribosomal complex, by cleaving the tRNA from the nascent chain, a function also performed by its mammalian homologue ANKZF1. Interestingly, ANKZF1 was shown to act specifically on ubiquitylated nascent chains [150]. Indeed, the RQC complex is also composed of the E3 ligase Ltn1 and the proteins Rqc1 and Rqc2 [142,143,147,151,152,153]. Ltn1 ubiquitylates the aberrant nascent peptides, which are escorted for proteasomal degradation by Cdc48 [154]. This is mediated by Rqc2 upon its recruitment to the peptides stalled in 60S ribosomal particles. There, Rqc2 recruits charged tRNAs, enabling elongation of the nascent protein with carboxy-terminal alanine and threonine extensions (CAT tails). The resulting mobilization of the nascent chain potentially allows exposure of lysine residues to be ubiquitylated by Ltn1. Finally, the RQC was shown to depend on Cdc48/p97 and its cofactors Ufd1 and Npl4, at least for cytosolic proteins [143,154,155].

In addition to its role in cytoRQC, Vms1 was also described to regulate ribosomes that are present at mitochondria (mitoRQC) [85] (Figure 3D). In fact, the vast majority of the mitochondrial proteome is translated in the cytoplasm, being posttranslationally directed for import into mitochondria. This occurs via recognition signals, mostly present in the protein N-terminus [26] but also present in the mRNA, thus leading to cotranslational insertion [156]. Vms1 is required to prevent aggregation and sequestration of nuclear-encoded mitochondrial proteins, present at ribosomes stalled at the mitochondrial surface [85]. The particular importance of Vms1 and RQC for mitochondrial functionality might be due to the fact that the nascent chain of mitochondrial proteins stalled on ribosomes can already expose a mitochondrial-targeting sequence. Consequently, defectively translated proteins, if inserted into mitochondria, can engage the mitochondrial import machinery and thereby escape cytosolic RQC, which strongly interferes with mitochondrial proteostasis [85]. However, the relationship between the hydrolase activity of Vms1 and its function as Cdc48 cofactor in each Vms1-dependent process remains to be elucidated. Furthermore, under mitoRQC, it is not known whether Vms1 is directly targeted to mitochondria via its MTD or indirectly by binding to the stalled ribosomes associated to the TOM channel.

### 3.4. Pro- and Antiapoptotic Roles of Cdc48 in Mitochondria

Cdc48/p97 was shown to have both antiapoptotic roles, e.g., by preventing ER stress and excessive oxidation, and proapoptotic roles, e.g., by facilitating cleavage and activation of caspases, a family of cysteine proteases [157,158] and the main effectors of apoptosis [82,159]. Under intrinsic cellular stress [10], caspases are activated by the release to the cytosol of several mitochondrial intermembrane space proteins, such as cytochrome c [160].

Under basal conditions, Cdc48/p97 targets numerous mitochondrial proapoptotic factors for degradation, preventing apoptosis [82,161]. Recently, such an antiapoptotic role of Cdc48 was described, involving the reduced nicotinamide adenine dinucleotide (NADH) dehydrogenase protein Nde1, the yeast homologue of the mammalian apoptosis-inducing factor (AIF) [162]. Nde1 forms two distinct topomers, residing in the IMS, ensuring enzymatic activity, or instead located at the OMM and facing the cytosol, triggering cell death. Under basal conditions, the surface-exposed topomer is degraded by the proteasome, dependently on Cdc48. However, under stress conditions, e.g., in respiratory-deficient cells, the surface-exposed topomer is stabilized, inducing selective elimination of these compromised cells from the population. Consistently, the mutant Cdc48^S565G^ leads to respiratory deficiency, accumulation of reactive oxygen species, and structural damage of mitochondria [88]. This results in cytochrome c translocation and apoptosis, emphasizing the importance of Cdc48 for apoptotic signalling. Along with an antiapoptotic role, p97 could also be connected to mitochondrial-associated Alzheimer disease (AD) [83]. Accumulation of ubiquitin B (UBB^+1^), a frame-shift, truncated form of ubiquitin leads to neuronal cell death and is a hallmark of Alzheimer disease. When expressed in yeast, UBB^+1^ disturbs the UPS, leading to mitochondrial stress and apoptosis. Overexpression of Cdc48 and Vms1, in yeast strains expressing UBB^+1^, downregulates apoptosis, by assisting proteolysis [83]. p97 was also shown to regulate calcium overload, which signals intrinsic apoptosis [124,163]. For example, in mouse hearts, p97 modulates mitochondrial proteins involved in calcium uptake, thereby protecting from Ca^2+^ overload [124]. Moreover, p97 regulates the inositol 1,4,5- trisphosphate receptor IP3R3 [163]. This receptor, which facilitates Ca^2+^ influx into mitochondria, is activated by the tumor-growth factor PTEN. IP3R3 is ubiquitylated by the SCF^FBXL2^ and targeted for proteasomal degradation, dependent on p97, damping Ca^2+^ influx and circumventing apoptotic signalling.

In contrast, p97 can also assist the turnover of antiapoptotic proteins. Here, a prominent target of p97 is the antiapoptotic protein Mcl1, an outer membrane protein of the Bcl-2 family, whose turnover enables apoptosis [71]. Upon induction of apoptosis, Mcl1 is rapidly ubiquitylated by several E3 ubiquitin ligases, namely HUWE1/ARF-BP1/MULE, SCF^FBW7^, and MARCH5 [71,164,165]. Subsequent membrane extraction of Mcl1 requires p97 and its cofactor UBXD1, allowing Mcl1 degradation by the proteasome, thus promoting apoptosis [166]. A different proapoptotic mechanism of p97 was described, associated to the control of mitochondrial dynamics. Expression of mutant hyperactive VCP in *Drosophila* muscle cells increased turnover of the fly mitofusin Marf [167]. The disturbance of mitochondrial plasticity thereof culminated in muscle damage and cell death. Interestingly, these mitochondrial defects could be suppressed by VCP inhibitors, in IBMPFD models of dementia.

### 3.5. Regulation of Mitophagy by Cdc48/p97-MAD of Mitofusins

p97 was suggested to regulate mitophagy, the selective autophagic clearance of mitochondria damaged by high stress levels, allowing segregation of damaged from healthy organelles, in order to maintain cellular integrity [168,169]. Mitophagy signalling is largely regulated by phosphorylation and ubiquitylation, mainly regulated by the PTEN-induced kinase 1 (PINK1) and the E3 ligase Parkin [14,15]. Upon mitochondrial stress, such as depolarization, PINK1 accumulates on the mitochondrial surface. Here, it phosphorylates Parkin and ubiquitin at serine 65, exponentially stimulating Parkin activity toward OMM proteins. This recruits autophagy receptors, ultimately leading to the engagement of the autophagic engulfment machinery [14,15]. p97 locates to mitochondria upon depolarization, thus enabling PINK1/Parkin-dependent mitophagy [58]. Studies in the fruit fly also showed VCP recruitment to damaged mitochondria, dependent on ubiquitylation of mitochondrial targets by Parkin [94]. In agreement with p97 acting downstream of PINK1/Parkin, p97 is not required for Parkin-mediated ubiquitylation of the mitochondrial proteome, in reprogrammed, induced neuron cells, i.e., with endogenous Pink1 and Parkin [170]. The recruitment of p97 to depolarized mitochondria was suggested to depend on the cofactor UBXD1 [171]. Moreover, silencing of Npl4, Ufd1, or p47 (Shp1 in yeast) impairs mitophagy [172], reinforcing the importance of p97.

Mitofusins are main regulators of mitochondrial integrity [68,69] and are probably the most extensively studied targets of Cdc48/p97 [71,132,133,173]. Originally, it was shown that Mfn1 and 2 are degraded upon mitochondrial depolarization [58]. This leads to p97 translocation to mitochondria, inducing mitophagy in a p97, Parkin, and proteasome-dependent manner [58]. Consistently, *Drosophila clueless* (*clu*) protein, which acts upstream of VCP, promotes VCP binding to the mitofusin Marf, assisting its turnover and triggering mitophagy [174]. Moreover, Fzo1 is degraded under stress conditions, dependent on Cdc48 [70,72]. Interestingly, in the abovementioned study using reprogrammed, induced neurons, mitofusins were the main substrates of endogenous PINK1 and Parkin, together with Miro, another prominent ubiquitin target, which modulates mitochondrial motility [170]. The degradation of mitofusins causes fragmentation of the mitochondrial network, due to ongoing fission events. It was proposed that inhibition of fusion facilitated segregation, engulfment, and mitophagy of the damaged mitochondrial fragments [58]. Other than fusion inhibition, it was also proposed that the turnover of phospho-ubiquitylated Mfn2 upon mitophagy induction, assisted by p97, reduces ER–mitochondria contacts [175]. In contrast, the presence of mitofusins at the OMM might be required for mitophagy [68,176]. Indeed, Mfn2 was proposed to be required for Parkin recruitment, thereby facilitating mitophagy [177]. Similarly, Mfn1 was proposed to be required for mitophagy, caused by overexpression of the E3 ligase Gp78 [178]. Therefore, the exact role of mitofusin turnover in mitophagy requires further investigations. Finally, upon mitochondrial depolarization, p97 assists the translocation of the E3 ligase MARCH5 from mitochondria to peroxisomes [179]. Together, these studies show the importance of p97/VCP for mitophagy.

Other than its putative relevance in Parkinson’s disease, associated to PINK1 or Parkin defects, the role of p97 in mitochondria was also suggested to be important for Huntington. One of the causes of Huntington disease is the accumulation of mutant Huntingtin (mtHtt) at mitochondria [180]. p97 was found to bind to mtHtt and thereby translocate to mitochondria, in a Huntington disease model [181]. This translocation was thought to cause excessive mitophagy, ultimately leading to neuronal cell death. Finally, the most common VCP pathogenic mutations showed reduced recruitment to depolarized mitochondria, pointing to another link between p97-associated disease and mitophagy [172].

## 4. Beyond MAD: Regulation of the Mitofusins’ Fusion Activity by Cdc48

Other than its role in stress-induced MAD of mitofusins, Cdc48/p97 regulates mitochondrial dynamics by stimulating the activity of Fzo1, thus activating mitochondrial fusion [68,72,74,75,76,132,133,182]. Generally, mitochondrial dynamics is mediated by large dynamin-like GTPase proteins (DRPs), a protein class whose function relies on self-oligomerization, coordinating GTP binding and hydrolysis with conformational changes [68,183,184,185,186]. In addition, their activity is fine-tuned by assembly with coregulatory proteins and by posttranslational modifications [187]. In regards to the OMM, fusion depends on ubiquitylation of mitofusins [188,189].

### 4.1. Mechanism Governing OMM Fusion

Mitofusins possess an N-terminal GTPase (G) domain, transmembrane (TM) domains, flanked by two heptad repeats (HR1 & 2) and a small C-terminal domain [190,191,192,193,194] (Figure 4A). The yeast, fly, and worm mitofusins contain an additional HR domain, N-terminally of the G domain (HRN), of unknown function, but essential for Fzo1 activity [188,195]. In vitro, the rate-limiting factor for fusogenic capacity was the accessibility of GTP, although the generally high cytosolic guanosine triphosphate/ diphosphate (GTP/GDP) ratio would suggest a more refined regulation [196,197].

While the precise mechanism allowing mitofusins to perform OMM fusion remains elusive, structural insights in the last years gave extremely valuable hints to elucidate the events leading to membrane merging [198,199,200,201]. Structures of the mitofusin homologue bacterial dynamin-like protein (BDLP) revealed two different conformations, depending on the nucleotide state [202,203]. A GMPPNP-bound dimer (mimicking the GTP-bound state) shows a stretched conformation [203], in contrast to a constricted dimer when bound to GDP [202] (Figure 4B). Based on these structures, truncated versions of Mfn1 and Mfn2 were designed, and termed minimal GTPase domain (MGD) (Figure 4A). MGDs allowed obtaining several monomeric and dimeric structures of Mfn1 and Mfn2 [198,199,200,201]. Strikingly, the dimer structures of Mfn1-MGD, either bound to nonhydrolysable GDP-BeF_4_^-^ or instead to GDP-AlF_3_^-^, recapitulate the stretched and constricted conformations of the BDLP dimer structures, respectively [199,200]. The structural information highlights drastic changes and reveals dimer interfaces and hinge points. Interestingly, these regions map to mutations associated with CMT2A [204], suggesting pathogenic relevance. Their importance as a driving force for membrane merging could be confirmed with subsequent functional analysis, allowing further dissection of the fusion mechanism [75,76,194,195,205,206,207,208,209,210,211,212].

Consistent with the self-assembly properties of DRPs, mitofusins can be found in multiple oligomerization states [184,207] (Figure 4C). First, mitofusins locate to mitochondria as GTP-bound dimers [213,214]. They insert into the OMM via their TM [195], assisted by helices from HR1 and HR2 [194,206]. Upon approximation of two mitochondria, mitofusins physically interact, allowing *trans* tethering to occur [213,214,215]. This depends on interactions between the GTPase domains (G–G interface) [199,200,201]. Subsequently, mitofusins undergo higher oligomerization [211,216]. Several possible assembly models have been proposed [207,208,209], likely involving interaction via the G–G interface and intermolecular hydrophobic interactions between the HR2 domains [195,217,218]. Although the exact triggers facilitating *trans* oligomerization are unknown, studies in mammals suggest that cytosolic signaling regulates Mfn2 oligomerization, e.g., via the oxidative state and the apoptosis regulator BAX [197,212,219,220]. The C-terminal HR domain was also suggested to assist *trans* oligomerization [217,218,221], a model under debate. Additionally, oligomerization promotes fusion by increasing membrane curvature [203,216]. Cryo-electron microscopy studies revealed Fzo1 complexes forming a compact tethering structure around the mitochondrial contact site, which expands upon GTP hydrolysis to a docking ring [216]. The docking ring likely extends the contact between mitochondria and simultaneously increases membrane curvature, ultimately leading to membrane merging [216].

Upon GTP hydrolysis, several conformational changes occur, which are critical to complete fusion. Bending at the hinge 1 constricts mitofusins, likely assisted by cytosolic factors [212]. This constriction is facilitated by dynamic changes at a trilateral salt bridge, present at hinge 2a [75,195,198]. Recently, a region surrounding HR1 was identified, conferring different fusogenic and assembly properties of Mfn1 and Mfn2 [210]. This mitofusin isoform-specific region (MISR) was proposed to facilitate oligomerization via conformational changes in hinge 1. Finally, remodeling of the G–G interface allows several structural rearrangements [76,199,200].

### 4.2. Roles of Ubiquitin and Cdc48 in OMM Fusion

Ubiquitylation of mitofusins is required for a tight regulation of its fusion activity [187,188,222]. Fzo1 was the first OMM protein identified to be regulated by ubiquitin, dependently and independently of the proteasome [159,223,224,225,226]. Then, the E3 ligases SCF^Mdm30^ and Parkin were shown to ubiquitylate yeast, human, and fly mitofusins [227,228,229]. Nowadays, many more E3 ligases have been identified [68]. Fzo1 ubiquitylation occurs at a late step in the fusion process, after *trans* oligomerization and GTP hydrolysis [214]. Deletion of the ligase SCF^Mdm30^ [214,224,225,226,230] or elimination of the lysines in Fzo1 required for its ubiquitylation [74,76,195,205] impairs fusion, leading to mitochondrial aggregation. These results defined that ubiquitylation is required for creating fusion-competent forms of Fzo1.

Other than mitochondrial aggregation, the absence of fusion-competent ubiquitylation leads to Fzo1 stabilization [224]. Initially, this suggested that excessive accumulation of Fzo1 would stall the tethering Fzo1 complexes, consistent with the aggregation of unfused mitochondrial fragments, which occur both upon deletion of *MDM30* [226] and upon Fzo1 overexpression [227]. Based on these observations, it was initially proposed that Fzo1 degradation is required for completion of mitochondrial fusion, to putatively release the *trans* complex from sterically hindering lipid merging [225,227]. However, subsequent studies identified two distinct forms of Fzo1 ubiquitylation either promoting or inhibiting mitochondrial fusion [205] and revealed that mitochondrial fusion depends on the amount of the profusion ubiquitylated forms conjugated to Fzo1, rather than on its steady-state levels [74,76,205] (Figure 5A). In fact, the profusion forms are composed of only very short K48-linked ubiquitin chains, which are not sufficient to target Fzo1 for degradation [74]. These highly conserved regulatory ubiquitin chains present an unusual pattern [228,229], which is essential for mitochondrial fusion [76]. On top of this conserved ubiquitin pattern, canonical ladder-like ubiquitin chains are formed on Fzo1, which address it for degradation by the proteasome, thereby destabilizing Fzo1 [74,205] (Figure 5A). This degradation signal, which leads to mitochondrial fragmentation, is important for metabolic adaptations [159,225,231] and to mitigate stress [135,232].

Cdc48 promotes mitochondrial fusion in several ways. It constitutively binds to the profusion ubiquitylation forms [74,132] and protects Fzo1 from further ubiquitylation and proteasomal degradation [74]. This raised the hypothesis that Cdc48 action on Fzo1 could be similar to the role of the AAA-ATPase N-ethylmaleimide sensitive fusion protein (NSF) in vesicle fusion, by disassembling SNARE complexes after completed membrane merging [233]. As mentioned above, the mitochondrial tethering complex consists of oligomers of at least three or four subunits, likely an even larger amount of Fzo1 molecules [207,214,216]. The conformational changes in Fzo1, induced by GTP hydrolysis, might allow the tethering complex to expand to the docking ring, leading to increased membrane tension in the narrow mitochondrial fusion side [216]. Furthermore, GTP hydrolysis leads to striking conformational changes in the G-domain itself, which allow Fzo1 ubiquitylation, Cdc48 recognition, and OMM fusion [76]. Prevention of Cdc48 binding to Fzo1 leads to the accumulation of Fzo1 at distinct foci at mitochondria, consistent with a role of Cdc48 in supporting membrane fusion by segregating Fzo1 complexes [75]. Also, in line with a segregation role of Cdc48, overexpression of Cdc48 resolves Fzo1 clusters that occur upon mutation of the small GTPase Arf1, a protein playing a critical role in membrane trafficking at mitochondria [173]. These observations suggest that Cdc48 could directly bind and pull Fzo1 ubiquitin chains into its catalytic pore. Instead of leading to proteasomal degradation, Fzo1 would be released from Cdc48 before complete unfolding of the protein, allowing disassembly and recycling of Fzo1 from the fusion complexes (Figure 5B). However, the exact mechanism via which Cdc48 directly binds and affects Fzo1 needs to be further elucidated. Indeed, ubiquitin chains of at least five moieties are required to achieve optimal binding of the Cdc48-Ufd1-Npl4 complex [103,234]. However, a length of only two ubiquitin moieties has been observed on the profusion chains of Fzo1 [74]. The observation that Cdc48-Shp1/Ubx1 can activate PP1 complexes, by segregating inactivating complex components, could suggest a similar role for Fzo1 [38]. However, opposed to PP1, recognition of Fzo1 by Cdc48 does require ubiquitylation [74,76] and a role of Shp1/Ubx1 for Fzo1 activity has not been described so far.

An additional mode of Cdc48 in preventing Fzo1 degradation involves the levels of Ubp2 and Ubp12 [74], the DUBs regulating Fzo1 ubiquitylation [205] (Figure 5C). The profusion ubiquitin forms are removed by Ubp12, whereas the proteasomal ubiquitin forms can be cleaved by Ubp2. These two DUBs, with opposing but sequential functions, form a deubiquitylase cascade, governed by Cdc48 [74]. Indeed, the activity of Ubp2 itself is repressed by Ubp12. This indicates a dual antifusion role of Ubp12, by removing profusion ubiquitin forms on Fzo1 and by promoting the accumulation of the prodegradative ubiquitin forms. Importantly, Cdc48 inhibits Ubp12 by binding to it and supporting proteasomal degradation of ubiquitylated Ubp12. In sum, in addition to its direct regulation of Fzo1 segregation, Cdc48 has a dual profusion function. Firstly, by downregulating Ubp12, thereby preserving regulatory profusion ubiquitin chains on Fzo1, and, secondly, by indirectly preserving Ubp2 function, thereby reducing proteasomal antifusion ubiquitylation on Fzo1. This highly sophisticated regulatory mechanism further suggests that both pro- and antifusion ubiquitin forms on Fzo1 are constantly attached and cleaved, allowing highly dynamic adaptations of Fzo1 activity and stability [74]. In contrast, the mammalian E3 ligase Mahogunin Ring Finger-1 (MGRN1) was shown to affect mitochondrial morphology by ubiquitylating Mfn1, which was proposed to allow its MAD by p97 and proteasomal degradation [235]. This discrepancy could be due to the specific Cdc48 cofactors acting on Fzo1/Mfn1 and their modulation of mitofusin ubiquitylation.

Two other Cdc48 interactors, Ubp3 and Ubx2, were recently shown to regulate mitochondrial fusion [132,133]. Deletion of *UPB3* leads to the stabilization of Ubp12, likely increasing the elimination of fusion-competent forms of Fzo1, consistent with the mitochondrial aggregation phenotype observed [132]. This would be compatible with a role of Ubp3 in removing degradative ubiquitylation from Ubp12 [132]. On the other hand, *UBX**2* deletion impairs complex dissociation, increases Fzo1 stabilization, and decreases Ubp2 stabilization, suggesting that Ubx2 stabilizes Ubp2 and, therefore, eliminates the degradative ubiquitylation forms on Fzo1 [132]. Indeed, Ubx2 regulation of Fzo1 was dependent on Ubp2, compatible with an increase of fusion-competent forms of Fzo1, due to the elimination of the antifusion forms by increased Ubp2 levels [133]. Moreover, Ubx2 physically interacts with Ubp2 and Fzo1, dependent on its UBX domain [133]. However, it remains to be determined if Ubx2 directly facilitates the recognition of Fzo1 complexes by Cdc48.

## 5. Conclusions

The importance of ubiquitin and Cdc48/p97 in mitochondrial proteostasis is increasingly clear. The last years allowed elucidating multiple housekeeping roles of Cdc48, in mitochondrial fusion, organelle identity and interconnection, protein import, and ribosomal quality control. So far, the pathways and players involved have been mainly discovered in baker’s yeast. Future studies will allow revealing conservation to higher organisms and pathophysiological implications. Moreover, increased stress loads engage Cdc48/p97 for whole organellar or cellular elimination, by mitophagy or apoptosis, respectively. In most cases, mitochondrial processes rely on the membrane extraction and unfolding capacity of Cdc48, allowing substrate delivery to proteasomal degradation. In addition, the novel nonproteolytic role of Cdc48 as a segregator of Fzo1 oligomers confirms the functional complexity of this AAA-ATPase, opening up great opportunities for further investigation.

## Figures and Tables

**Figure 1 ijms-21-06841-f001:**
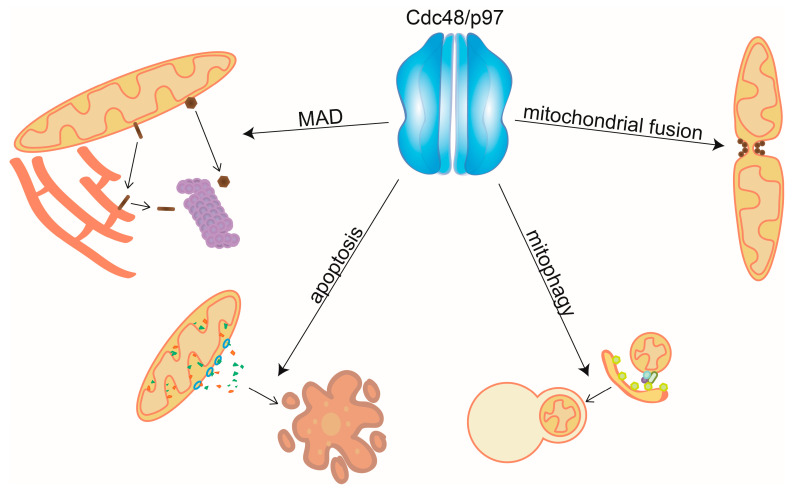
Mitochondrial functions of Cdc48/p97. Cdc48/p97 mediates mitochondrial-associated degradation (MAD), by extraction and unfolding of mitochondrial substrates and promotes mitochondrial fusion by segregating and disassembling the fusion complex. By assisting the degradation of both pro- and antiapoptotic factors, Cdc48/p97 regulates apoptosis and by controlling the ubiquitylation levels at mitochondria it enables PINK1/Parkin dependent mitophagy.

**Figure 2 ijms-21-06841-f002:**
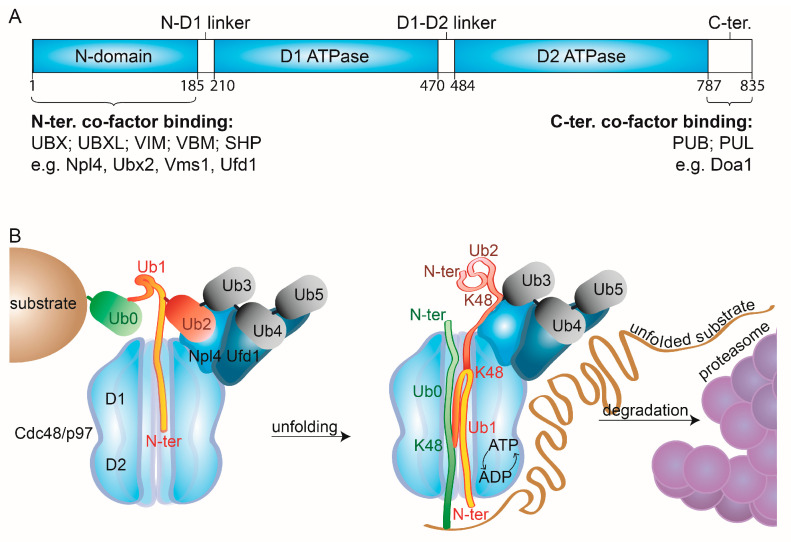
Cdc48/p97 mechanistic features. (**A**) Linear representation of the domain structure of Cdc48/p97. (**B**) Molecular modes of action of Cdc48/p97. Cdc48 and its cofactors, Ufd1 and Npl4, bind the substrate’s ubiquitin chains. Structural analysis shows two distal ubiquitin moieties (Ub_4_ & Ub_5_) recognized by Ufd1, two precedent moieties (Ub_2_ & Ub_3_) by Npl4, and the precedent moiety (Ub_1_) unfolded and inserted into the catalytic pore of Cdc48, via its N-domain [100]. Subsequently, ATP hydrolysis and ADP-ATP shuffling allow unfolding of additional ubiquitin moieties (Ub_0_) and of the substrate, enabling its degradation by the proteasome, while Ub_3_ to Ub_5_ remain folded. N-domain, amino-terminal domain; D1&2, ATPase domain 1&2; N-ter, amino-terminus, C-ter, carboxy-terminus.

**Figure 3 ijms-21-06841-f003:**
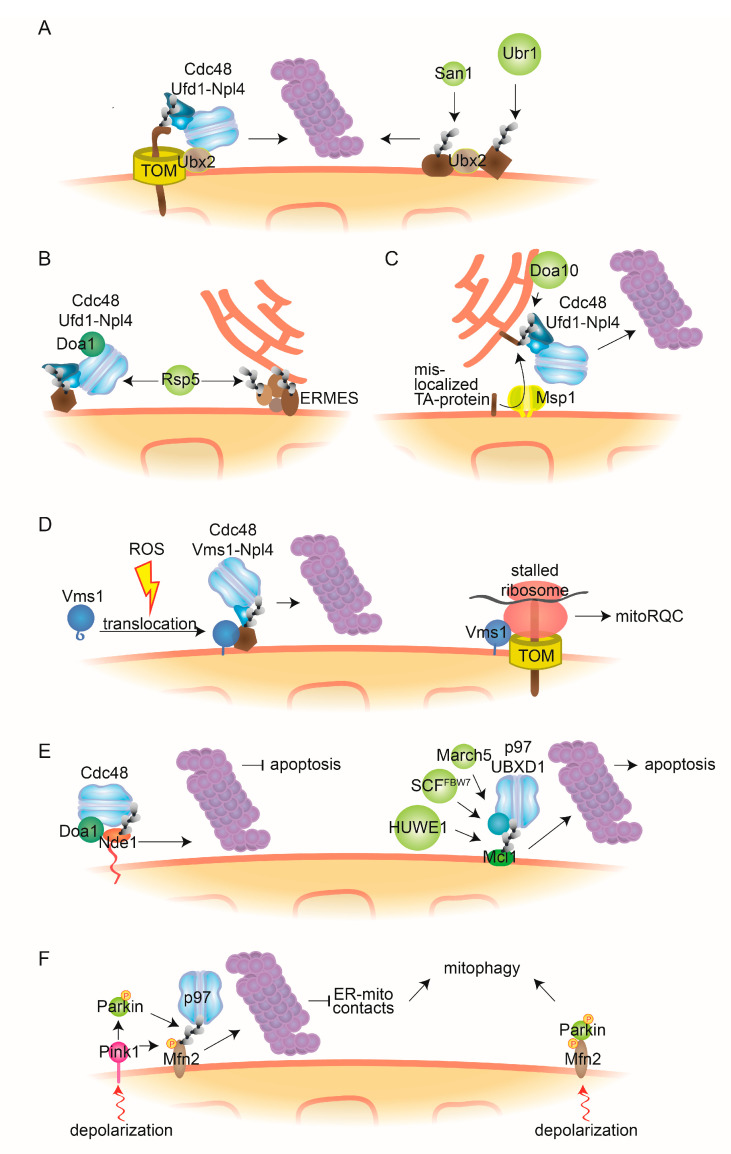
Cdc48-dependent MAD and mitochondrial quality control mechanisms. (**A**) Ubx2-dependent MAD. The Cdc48-cofactor Ubx2 partially localizes to the OMM where it facilitates the degradation of ubiquitylated mitochondrial proteins bound to the TOM channel, dependent on the Cdc48-Ufd1-Npl4 complex. Further, Ubx2 recognizes mitochondrial substrates that are ubiquitylated by the quality control E3 ligases Ubr1 and San1. (**B**) Doa1-dependent MAD. The Cdc48-Ufd1-Npl4 complex, together with the cofactor Doa1/Ufd3, assists proteasomal degradation of OMM proteins ubiquitylated by E3 ligase Rsp5. Additionally, Rsp5 can ubiquitylate the ERMES components Mdm12 and Mdm34. (**C**) ERAD-assisted degradation. The AAA-ATPase Msp1 facilitates the retro-translocation to the ER of TA-proteins mistargeted to the OMM. After ubiquitylation by the ER-bound E3 ligase Doa10, Cdc48-Ufd1-Npl4 engages the canonical ERAD pathway. (**D**) Vms1-dependent MAD. Oxidative stress, such as reactive oxygen species (ROS), enables translocation to mitochondria of the Cdc48 cofactor Vms1, where it recruits Cdc48 and Npl4 to assist proteasomal degradation of ubiquitylated mitochondrial substrates. Additionally, Vms1 recognizes stalled ribosomes associated to TOM channels and facilitates release of aberrant nascent peptides, by mitoRQC. (**E**) Cdc48/p97 in apoptosis. Accumulation of the cytosol-exposed topomer of Nde1 leads to induction of mitophagy. Cdc48 recognizes ubiquitylated forms of this topomer, dependent on Doa1, and facilitates its degradation, thereby preventing apoptosis (left). p97 and its cofactor UBXD1 facilitate proteasomal turnover of the antiapoptotic protein Mcl1, ubiquitylated by the E3 ligases HUWE1, SCF^FBW7^ or MARCH5, thereby inducing apoptosis (right). (**F**) p97 in mitophagy. Upon mitochondrial depolarization, PINK1 accumulates at the OMM, where it phosphorylates Parkin, leading to its activation. Parkin-mediated ubiquitylation and p97-mediated degradation of Mfn2 induces mitophagy, e.g., via reduction of ER-mitochondria contacts (left). Alternatively, phosphorylated Mfn2 serves as an adaptor for Parkin, enabling mitophagic signalling (right). MitoRQC, mitochondrial ribosomal quality control; ERMES, ER-mitochondria encounter structure; TOM, translocase of the outer mitochondrial membrane complex.

**Figure 4 ijms-21-06841-f004:**
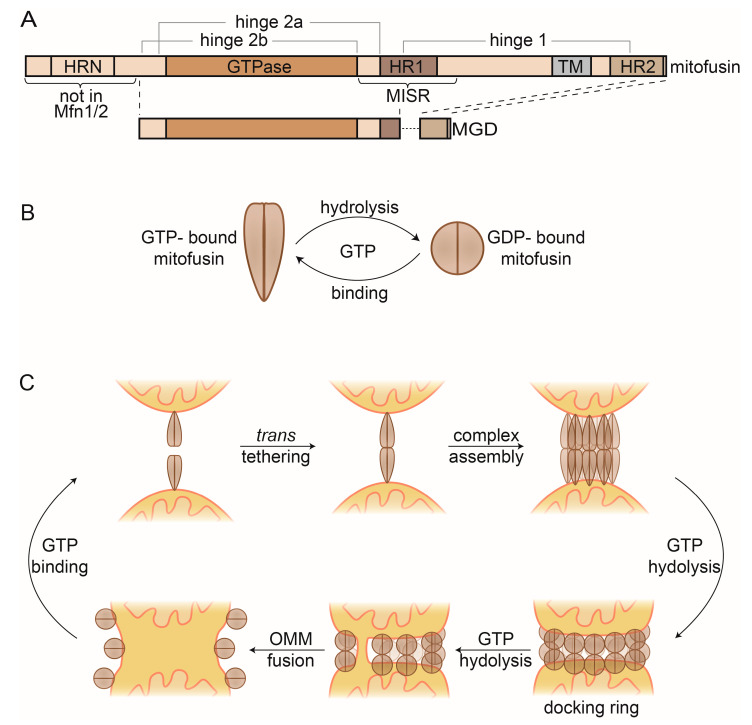
Mitofusins and OMM fusion. (**A**) Linear representation of the domain structure of full-length mitofusins and the minimal GTPase domain (MGD). (**B**) GTP-dependent conformational changes of mitofusin dimers. GTP-bound mitofusin is in a stretched conformation, which constricts upon GTP hydrolysis. (**C**) Current model of OMM fusion. GTP-bound Fzo1 dimers on the OMM facilitate mitochondrial tethering via *trans* oligomerization. After higher complex formation, GTP hydrolysis occurs, inducing conformational changes that mediate formation of the docking ring. Repeated rounds of GTP binding and hydrolysis presumably lead to localized membrane merging that then expands for complete outer membrane fusion. Finally, mitofusin complexes are disassembled and recycled, via GTP binding. HR, heptad repeat; MISR, mitofusin isoform-specific region.

**Figure 5 ijms-21-06841-f005:**
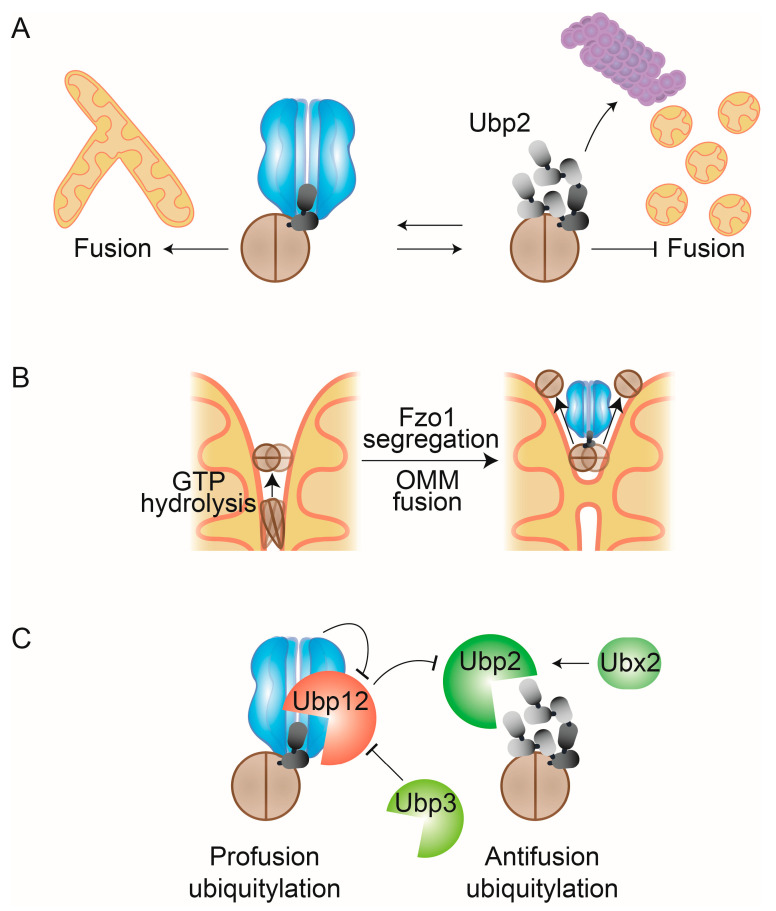
Regulation of OMM fusion by Cdc48. (**A**) After GTP hydrolysis, Fzo1 is modified by short ubiquitin chains that are recognized by Cdc48 and that promote mitochondrial fusion. Further ubiquitylation of Fzo1creates a degradation signal that instead targets Fzo1 for degradation by the proteasome, thereby inhibiting fusion. (**B**) Cdc48 enables Fzo1-dependent membrane merging. OMM merging requires relocalization of mitofusins away from the site of membrane merging, likely mediated by conformational changes of Fzo1, dependent on GTP-hydrolysis. After GTP-hydrolysis Fzo1 can be ubiquitylated and recognized by Cdc48. This leads to complex segregation, allowing OMM fusion and recycling of Fzo1. (**C**) Cdc48-dependent DUB cascade. Profusion ubiquitylated Fzo1 forms are cleaved by the DUB Ubp12, while antifusion forms, which are built on previously attached regulatory forms, are cleaved by the DUB Ubp2. Cdc48 inhibits Ubp12 activity, while Ubp12 inhibits Ubp2 activity. Thereby, Cdc48 promotes fusion by firstly inhibiting cleavage of profusion ubiquitin forms on Fzo1 and secondly by promoting cleavage of proteolytic ubiquitin forms on Fzo1. Furthermore, Ubp12 is negatively regulated by the DUB Ubp3 while Ubp2 is stabilized by Ubx2.

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
