# Peer review of "Mitochondrial Surveillance by Cdc48/p97: MAD vs. Membrane Fusion"

_ijms, 2020, doi:10.3390/ijms21186841_

Round 1

Reviewer 1 Report

This is a very comprehensive review on the multiple roles of Cdc48/p97 in mitochondrial homeostasis, signaling and disease. Even though several related reviews appeared recently (e.g., by Johannes Herrmann in TCB, Bingwei Lu in TCB, Thorsten Hoppe in Frontiers in Cell Dev Biol, and by the corresponding author herself in Curr Issue Mol Biol), the main focus on Cdc48 and its control of mitofusins has not been covered in such depth before and will stimulate readers from the ubiquitin and mitochondria fields to think out of the box.

General comment:

1. As mentioned, the focus on the interactions between Cdc48 and mitofusins is unique compared to similar reviews, and should therefore be the in the center of the review. The authors cover also many other aspects of Cdc48-mitochondria interactions, but in several places do so in a slightly superficial way lacking functional or mechanistic insights. E.g., the part on apoptosis signaling cut be shortened or omitted. A more straightforward organization of the manuscript could be to introduce the concept of Cdc48-mediated MAD, mention several examples, and then focus entirely on mitofusins and the (in part controversial) roles that Cdc48 plays in pro- versus anti-fusion form turnover and stress-induced degradation as prerequisite to mitophagy.

Minor general points:

2. It would be helpful to mention right at the beginning that the review generally focuses on the yeast system, with some comparisons to metazoan systems. Regarding the protein nomenclature, it is not always clear if a name refers to a yeast or metazoan protein.

3. While the authors did a tremendous job in collecting a large number of references, the citation logic is not always consistent. Mostly, the authors cite original papers, but sometimes they appear to take a shortcut and cite reviews where readers would prefer to get to know the original papers.

4. The English language and style need thorough editing.

Specific comments:

5. Introduction: A bit unbalanced – the UPS and Cdc48/p97 get 25 lines, followed by 50 lines about mitochondria (could be shortened), before few more lines about Cdc48 and MAD follow. Why not start with mitochondria and then move on to UPS, Cdc48 and MAD?

Regarding Cdc48, the authors state that this is a ubiquitin-specific ATPase. However, Hemmo Meyer´s lab has shown recently that the p97 cofactors p47, p37 and UBXN2A (homologues of Shp1/Ubx1) are involved in the ubiquitin-independent maturation of protein phosphatase 1 complexes.

Lines 108/109 and 114/115 are redundant.

6. Section 2: Fig. 2B – How can the substrate be completely threaded and unfolded, if Ub1 and Ub0 are still stuck in the central pore? In the legend, "pooled" should read "pulled" (line 128; also in line 179), and ref. 93 should read ref. 94 (line 129).

Line 173: The Cdc48-Shp1/Ubx1 structure (ref 93) is mentioned, but never described or discussed. What do we learn from that structure?

Line 185/186: Proteasomal ATPase possess only one ATPase ring, making this sentence confusing.

7. Section 3: Fig. 3 – For clarity, all Cdc48 cofactors should be marked and labeled (including Ufd1-Npl4). In 2B, it appears that Ubx2 acts downstream of Cdc48 – why? In the legend (and main text) several permutations of the established nomenclature "Cdc48-Ufd1-Npl4" are used.

Lines 242-279: This part lacks clarity. Why are different cofactors required to recruit Cdc48 to the OMM? Why do different substrates require different cofactors?

Re Vms1 (refers also to Section 3.3): How can Vms1 recruit stalled ribosomes to the OMM for mitoRQC when its MTD is masked (abscence of oxidative stress)? The discussion of Vms1 is slightly unbalanced towards mitoRQC, as more evidence has been published on a role of Vms1/ANKZF1 in cytoRQC. (Note that ref. 78 does not provide evidence for mitoRQC.) For completeness, the paper by Kuroha et al. (Mol Cell 2018) on ANKZF1 should be mentioned.

8. Section 3.1: The Cdc48/p97-dependent proteasomal degradation of mitofusins is an early and essential event in mitophagy. However, it mechanistically resembles Cdc48-mediated MAD of mitofusins. The authors should also note that different mechanisms and functions have been proposed for the degradation of Mfn1 versus Mfn2. Is there any solid evidence that Cdc48/p97 has additonal roles in mitophagy? If not, this section could be skipped and integrated into a discussion of Cdc48-mediated turnover of mitofusins.

9. Section 3.2: Could be skipped, as no clear picture emerges for the mechanisms and biology of pro-/anti-apoptotic functions of Cdc48/p97.

10. Section 4.2: While the discussion of Cdc48´s role in turnover of pre- versus anti-fusion ubiquitinylated Fzo1 is highly interesting, the paragraphs about the involvement of cofactors and DUBs are confusing (lines 516-546). The authors state that Ufd1-Npl4 is an unlikely candidate for the turnover of oligo-ubiquitinylated Fzo1 – what about Shp1/Ubx1, which exerts similar functions in PP1 maturation (see point 5 above)? What role do the authors suggest for Ubx2?

The logic behind the Ubp12-Ubp2 DUB cascade is unclear: In Fig. 4 it is stated that the anti-fusion/pro-degradation ubiquitin chain is built on pre-attached pro-fusion ubiquitin moieties. If Ubp12 removes the pro-fusion moieties, how can it at the same time promote the accumulation of pro-degradative chains (which are assembled on pro-fusion Ub)? Also, how does Cdc48 exactly down-regulate Ubp12, and why does deletion of UBP3 counteract this?

The authors should try to integrate all Cdc48 cofactors and DUBs into their model in Fig. 4D.

Author Response

Reviewer 1

“This is a very comprehensive review on the multiple roles of Cdc48/p97 in mitochondrial homeostasis, signaling and disease. Even though several related reviews appeared recently (e.g., by Johannes Herrmann in TCB, Bingwei Lu in TCB, Thorsten Hoppe in Frontiers in Cell Dev Biol, and by the corresponding author herself in Curr Issue Mol Biol), the main focus on Cdc48 and its control of mitofusins has not been covered in such depth before and will stimulate readers from the ubiquitin and mitochondria fields to think out of the box.”

We are thankful for the thorough review and have improved the structure and content of the manuscript accordingly. The order of topics has been changed in the introduction, first describing mitochondria. Further, we extended the description regarding the mechanism of Cdc48 substrate translocation. Chapter 3 was restructured to better emphasize the different roles of Cdc48/p97 in MAD in mitochondria. Therefore, Figure 3 has been changed accordingly. Lastly, chapter 4 Figure 4D have been modified as suggested.

“1. As mentioned, the focus on the interactions between Cdc48 and mitofusins is unique compared to similar reviews, and should therefore be the in the center of the review. The authors cover also many other aspects of Cdc48-mitochondria interactions, but in several places do so in a slightly superficial way lacking functional or mechanistic insights. E.g., the part on apoptosis signaling cut be shortened or omitted. A more straightforward organization of the manuscript could be to introduce the concept of Cdc48-mediated MAD, mention several examples, and then focus entirely on mitofusins and the (in part controversial) roles that Cdc48 plays in pro- versus anti-fusion form turnover and stress-induced degradation as prerequisite to mitophagy.“

The introduction and chapter 3 have been extensively re-organized, including mechanistic illustrations of both apoptosis and mitophagy (see new Fig. 3 E and F, respectively). Given that our review aims to give an overview of all described roles in mitochondrial regulation by Cdc48/p97, we would prefer not to omit the apoptosis and mitophagy chapters.

“2. It would be helpful to mention right at the beginning that the review generally focuses on the yeast system, with some comparisons to metazoan systems. Regarding the protein nomenclature, it is not always clear if a name refers to a yeast or metazoan protein.“

We are aware of this issue and thus have addressed it in chapter 5 (see lines 591-600). We have covered the available literature which was indeed mostly concerned discovers in yeast. However, we did not intentionally focus on yeast.

“3. While the authors did a tremendous job in collecting a large number of references, the citation logic is not always consistent. Mostly, the authors cite original papers, but sometimes they appear to take a shortcut and cite reviews where readers would prefer to get to know the original papers.”

Original references have been added in chapter 2. To the best of our knowledge, we carefully looked for the first time the findings we cite here were discovered, having added current and appropriate reviews whenever we described a general concept.

“4. The English language and style need thorough editing.”

The English language and style have been revised.

“5. Introduction: A bit unbalanced – the UPS and Cdc48/p97 get 25 lines, followed by 50 lines about mitochondria (could be shortened), before few more lines about Cdc48 and MAD follow. Why not start with mitochondria and then move on to UPS, Cdc48 and MAD?”

We thank Reviewer 1 for this very helpful comment.

“Regarding Cdc48, the authors state that this is a ubiquitin-specific ATPase. However, Hemmo Meyer´s lab has shown recently that the p97 cofactors p47, p37 and UBXN2A (homologues of Shp1/Ubx1) are involved in the ubiquitin-independent maturation of protein phosphatase 1 complexes.”

This aspect is indeed very important and is now discussed in lines 74-76 of the introduction. The statement “ubiquitin-specific” has been removed from the manuscript.

„Lines 108/109 and 114/115 are redundant.“

Indeed! The sentence previously present in lines 114/115 has been removed.

“6. Section 2: Fig. 2B – How can the substrate be completely threaded and unfolded, if Ub1 and Ub0 are still stuck in the central pore?”

This idea is based on the observation that translocation initially happens by unfolding of a proximal ubiquitin, Ub1 in this case. Pulling of the N-terminal end of ubiquitin would lead to subsequent pulling and degradation of the surrounding ubiquitin moieties, Ub0 and Ub2 in this case, and the substrate, however leaving the distal ubiquitin moieties, Ub3,4,5 in this case, in their native state, This implies that ubiquitin moieties remain in the central pore of Cdc48, even after substrate unfolding. We acknowledge that it is not absolutely clear in what order and exact configuration the unfolding of the substrate and the attached ubiquitins happens. The corresponding text passage has been extended to clarify and discuss this model (lines 184-199).

“In the legend, "pooled" should read "pulled" (line 128; also in line 179), and ref. 93 should read ref. 94 (line 129)”

The text passages have been changed. The reference has been corrected and is now ref. 100 (see line 152).

“Line 173: The Cdc48-Shp1/Ubx1 structure (ref 93) is mentioned, but never described or discussed. What do we learn from that structure?“

The text passage referring to ref 93 (new ref. 99) has been extended to further highlight this contribution (see lines 200-213).

“Line 185/186: Proteasomal ATPase possess only one ATPase ring, making this sentence confusing.”

The corresponding sentence has been changed (see lines 200-203).

“7. Section 3: Fig. 3 – For clarity, all Cdc48 cofactors should be marked and labelled (including Ufd1-Npl4).”

Labelling of all co-factors has been added.

“In 2B, it appears that Ubx2 acts downstream of Cdc48 – why?”

This figure (new Fig. 2 A) has been changed to avoid the impression that Ubx2 acts downstream of Cdc48.

“In the legend (and main text) several permutations of the established nomenclature "Cdc48-Ufd1-Npl4" are used.”

The nomenclature of “Cdc48-Ufd1-Npl4” has now been consistently used throughout the text.

“Lines 242-279: This part lacks clarity. Why are different cofactors required to recruit Cdc48 to the OMM? Why do different substrates require different cofactors?“

An explanatory text passage has been added to explain the general idea that substrate specificity and adaptation to cellular condition of Cdc48 is determined by its co-factos (lines 227-229).

“Re Vms1 (refers also to Section 3.3): How can Vms1 recruit stalled ribosomes to the OMM for mitoRQC when its MTD is masked (absence of oxidative stress)? The discussion of Vms1 is slightly unbalanced towards mitoRQC, as more evidence has been published on a role of Vms1/ANKZF1 in cytoRQC. (Note that ref. 78 does not provide evidence for mitoRQC.) For completeness, the paper by Kuroha et al. (Mol Cell 2018) on ANKZF1 should be mentioned.“

This text has been remodelled accordingly and mitochondrial targeting of Vms1 in RQC is now discussed (see lines 335-337). Reference 78 (new ref. 86) has been eliminated from this point. The paper by Kuroha et. al. is now included as reference 151 (see line 310).

“8. Section 3.1: The Cdc48/p97-dependent proteasomal degradation of mitofusins is an early and essential event in mitophagy. However, it mechanistically resembles Cdc48-mediated MAD of mitofusins. The authors should also note that different mechanisms and functions have been proposed for the degradation of Mfn1 versus Mfn2. Is there any solid evidence that Cdc48/p97 has additonal roles in mitophagy? If not, this section could be skipped and integrated into a discussion of Cdc48-mediated turnover of mitofusins.”

As mentioned above, chapter 3 has been extensively revised. The specific role of mitofusin and its degradation in mitophagy is now described in chapter 3.5 and depicted in Fig. 3 F.

“9. Section 3.2: Could be skipped, as no clear picture emerges for the mechanisms and biology of pro-/anti-apoptotic functions of Cdc48/p97.“

Similar to point 8, this section (now 3.4) has been revised and illustrated in Fig. 3 E.

“10. Section 4.2: While the discussion of Cdc48´s role in turnover of pre- versus anti-fusion ubiquitinylated Fzo1 is highly interesting, the paragraphs about the involvement of cofactors and DUBs are confusing (lines 516-546). The authors state that Ufd1-Npl4 is an unlikely candidate for the turnover of oligo-ubiquitinylated Fzo1 – what about Shp1/Ubx1, which exerts similar functions in PP1 maturation (see point 5 above)? What role do the authors suggest for Ubx2?“

We thank the reviewer for this comment. The discussion of the possible role of Ubx2 has been extended (see lines 571-582) and a possible role of Shp1/Ubx1 is now discussed (see lines 554-558)

“The logic behind the Ubp12-Ubp2 DUB cascade is unclear: In Fig. 4 it is stated that the anti-fusion/pro-degradation ubiquitin chain is built on pre-attached pro-fusion ubiquitin moieties. If Ubp12 removes the pro-fusion moieties, how can it at the same time promote the accumulation of pro-degradative chains (which are assembled on pro-fusion Ub)? Also, how does Cdc48 exactly down-regulate Ubp12, and why does deletion of UBP3 counteract this?”

The Ubp12/Ubp2 DUB cascade acting on Fzo1 has been revised (lines 559–572). The possible role of Ubp3 is discussed in lines 577-581.

“The authors should try to integrate all Cdc48 cofactors and DUBs into their model in Fig. 4D.”

Fig 4D has been revised accordingly.

Reviewer 2 Report

This manuscript by Henriques and Anton reviews the role and mechanisms of Cdc48/p97 in mitochondrial associated degradation (MAD). The content is interesting, the paper is well written, and the science behind it has been previously validated through peer-review. I only have a few recommendations that should be clarified before publication:

Minor Comments:

Line 67: Finally, mitochondria play an essential role in buffering Ca2+ overload from the ER. Please add “from the ER and cytoplasm”.

Line 68: Mis-regulation of calcium homeostasis is commonly associated with metabolic dysfunctions such as type 2 diabetes. Please add associated with many diseases including metabolic dysfunctions such as type 2 diabetes, and neurodegeneration.

Line 70: By serving as a cell death gatekeeper, please mention how mitochondria involve in cell death such as mPTP opening or apoptosis.

Line 82: the quality of the mitochondrial proteome is regulated both by internal proteases and by the ubiquitin-proteasome system (UPS). Please add chaperone as they have an important role in mitochondrial proteome regulation.

References should be checked carefully. Some of the important references are either missing or cited without journal name. Please correct/add references.
Few examples are here.
Ref no. 35 - there is no name of journal - Ravanelli, S.; den Brave, F.; Hoppe, T. Mitochondrial Quality Control Governed by Ubiquitinn article. 646 2020, 10.3389/fcell.2020.00270/full, doi:10.3389/fcell.2020.00270/full.
Article from Aaron Ciechanover group showing ubiquitination of matrix protiens. PMID: 27157140; “Ubiquitination of specific mitochondrial matrix proteins and Giovanni Benard's report on ubiquitin mediated protein quality control of SDHA. PMID: 29874573; “Ubiquitin-Dependent Degradation of Mitochondrial Proteins Regulates Energy Metabolism” can be added.

There are several grammatical mistakes throughout the manuscript. The authors should have a native English speaker to review the manuscript before resubmission.

Author Response

Reviewer 2

“This manuscript by Henriques and Anton reviews the role and mechanisms of Cdc48/p97 in mitochondrial associated degradation (MAD). The content is interesting, the paper is well written, and the science behind it has been previously validated through peer-review. I only have a few recommendations that should be clarified before publication: “

We are pleased that the reviewer appreciated our manuscript and wish to thank the reviewer for the valid recommendations.

Minor Comments:

Line 67: Finally, mitochondria play an essential role in buffering Ca2+ overload from the ER. Please add “from the ER and cytoplasm”

The text has been changed accordingly (see new lines 39-40).

“Line 68: Mis-regulation of calcium homeostasis is commonly associated with metabolic dysfunctions such as type 2 diabetes. Please add associated with many diseases including metabolic dysfunctions such as type 2 diabetes, and neurodegeneration.”

Neurodegeneration is now referenced (see new lines 40-42 and reference 9).

“Line 70: By serving as a cell death gatekeeper, please mention how mitochondria involve in cell death such as mPTP opening or apoptosis.”

The involvement of mitochondria in cell death has now been further elucidated, in lines 43-46.

“Line 82: the quality of the mitochondrial proteome is regulated both by internal proteases and by the ubiquitin-proteasome system (UPS). Please add chaperone as they have an important role in mitochondrial proteome regulation.”

This modification has been included (see new lines 64-66).

“References should be checked carefully. Some of the important references are either missing or cited without journal name. Please correct/add references.

Few examples are here.

Ref no. 35 - there is no name of journal - Ravanelli, S.; den Brave, F.; Hoppe, T. Mitochondrial Quality Control Governed by Ubiquitinn article. 646 2020, 10.3389/fcell.2020.00270/full, doi:10.3389/fcell.2020.00270/full.”

References have been corrected.

“Article from Aaron Ciechanover group showing ubiquitination of matrix protiens. PMID: 27157140; “Ubiquitination of specific mitochondrial matrix proteins and Giovanni Benard's report on ubiquitin mediated protein quality control of SDHA. PMID: 29874573; “Ubiquitin-Dependent Degradation of Mitochondrial Proteins Regulates Energy Metabolism” can be added.”

We acknowledge that many references on ubiquitin in mitochondria have not been mentioned. However, we would like to stress that our study focusses on the roles of Cdc48 in mitochondria, not in the general roles of ubiquitin in mitochondria. This is now clearly stated in the abstract. In fact, here, we wished to avoid redundancies with other reviews available on that topic, including a recent book chapter from our group.

 “There are several grammatical mistakes throughout the manuscript. The authors should have a native English speaker to review the manuscript before resubmission.”

The English language has been revised.

Reviewer 3 Report

The authors provide a comprehensive, well-documented and nicely illustrated study about the role of Cdc48/p97 in regulation of mitochondrial proteome, well done.

Please check "Huntingtin's" vs "Huntington's" in lines 321-323.

Author Response

Reviewer 3

“The authors provide a comprehensive, well-documented and nicely illustrated study about the role of Cdc48/p97 in regulation of mitochondrial proteome, well done. Please check "Huntingtin's" vs "Huntington's" in lines 321-323.”

We are very thankful for the reviewer’s time and for these compliments. The mistake has been corrected (see new lines 419-421).